# Altered MANF Expression in Pancreatic Acinar and Ductal Cells in Chronic Alcoholic Pancreatitis: A Cross-Sectional Study

**DOI:** 10.3390/biomedicines11020434

**Published:** 2023-02-02

**Authors:** Nicholas J. Caldwell, Hui Li, Andrew M. Bellizzi, Jia Luo

**Affiliations:** 1Department of Pathology, Carver College of Medicine, University of Iowa, Iowa City, IA 52242, USA; 2Department of Pathology, Massachusetts General Hospital, Boston, MA 02114, USA; 3Iowa City VA Health Care System, Iowa City, IA 52246, USA

**Keywords:** alcohol abuse, acinar cells, ductal cells, endoplasmic reticulum stress, mesencephalic astrocyte-derived neurotrophic factor, pancreatitis

## Abstract

Background: Mesencephalic astrocyte-derived neurotrophic factor (MANF) is an endoplasmic reticulum (ER) stress response protein that plays an important role in pancreatic functions. As both alcohol and ER stress response proteins are involved in the pathogenesis of pancreatitis, we sought to investigate the expression of MANF in chronic alcoholic pancreatitis (CAP) and chronic non-alcoholic pancreatitis (CNP). Methods: A cohort of chronic pancreatitis tissues was gathered from routine surgical pathology (*n* = 77) and autopsy (*n* = 10) cases and tissue microarrays were created. Sampled tissues were reviewed and designated as representing CAP (*n* = 15), CNP (*n* = 58), or normal pancreatic tissue (NPT) (*n* = 27). MANF immunohistochemistry (IHC) and digital image analysis were performed to obtain an estimation of tissue fibrosis and an optical density (OD) of MANF IHC in ducts and acini for each case. The averaged values for these variables among histologic designations were compared. Results: The amount of fibrous tissue of the combined CAP and CNP group (chronic alcoholic and non-alcoholic pancreatitis, CANP) exceeded that of the NPT group (70% vs. 34%, *p* < 0.0001). The MANF OD in ducts of CANP was significantly higher than that of NPT (0.19 vs. 0.10, *p* < 0.05). The MANF OD in ducts of CAP was significantly higher than that of CNP (0.27 vs. 0.17, *p* < 0.05). The MANF OD in acini of CAP was significantly lower than that in CNP (0.81 vs. 1.05, *p* < 0.05). Finally, there was a statistically significant positive relationship between the amount of fibrosis and MANF OD in ducts (*p* < 0.001). Conclusions: MANF expression was higher in ducts of CAP than CNP. In contrast, MANF expression in acini was lower in CAP than CNP and NPT. There was a positive correlation between fibrosis and MANF levels in the ducts.

## 1. Introduction

Chronic pancreatitis (CP) is a progressive inflammatory disorder of the pancreas that adversely affects quality of life due to irreversible injury to the pancreas. The pathologic features of CP are well established and include acinar injury, ductal strictures and distortion, dysfunction of exocrine and endocrine pancreatic glands, and pancreatic atrophy and fibrosis [1]. Alcohol abuse is the most common risk factor for CP and is associated with 40–70% of all CP cases [2,3,4]. The pathogenesis of alcoholic CP remains incompletely understood. Only a small percentage of heavy drinkers will develop clinical and histologic features of CP [5,6,7]. In cellular and animal models, alcohol exposure alone fails to fully recapitulate the sequelae of pancreatitis [8,9,10,11]. This suggests that additional factors account for an individual’s differing susceptibility to the toxic effect of alcohol on the pancreas.

The endoplasmic reticulum (ER) is an intracellular compartment that plays a major role in protein folding and processing as well as calcium storage and release, and it also serves as the first step of the secretory pathway followed by the Golgi apparatus [12]. The function of the ER requires optimal protein processing, redox conditions, and ion concentrations such as calcium levels. The disruption of these processes results in the accumulation of unfolded or misfolded proteins in the ER lumen, which is a known ER stress and triggers an adaptive response known as unfolded protein response (UPR). The UPR is regulated by three transmembrane ER signaling proteins: pancreatic endoplasmic reticulum kinase (PERK), inositol-requiring enzyme 1 (IRE1), and activating transcription factor 6 (ATF6). UPR is activated to alleviate ER stress; however, when ER stress is severe and sustained and UPR is unable to restore ER homeostasis, cell death occurs. Mesencephalic astrocyte-derived neurotrophic factor (MANF) is an ER-stress-inducible protein that has been shown to play an important role in regulating the survival and proliferation of pancreatic beta cells [13,14]. MANF was originally identified as a neurotrophic factor for midbrain dopamine neurons and later was shown to play a role in regulating ER stress and UPR in a variety of disease models [11,15,16,17]. Pancreatic acinar cells, which are the majority epithelial cell type in the pancreas and hypothesized to be the initiation site of pancreatitis, are particularly vulnerable to ER dysfunction because of their dependence on ER functionality to produce sufficient digestive enzymes [10,18,19]. Disorders in ER function are associated with both alcoholic and non-alcoholic CP [11,20]. ER stress and the activation of UPR signaling have been shown in a variety of experimental models of alcohol-induced pancreatic damages [11]. MANF is upregulated in response to a number of ER stress inducers, including ethanol, and to exert beneficial effects in multiple cellular and animal models [21,22,23,24]. Previously, we showed that alcohol exposure causes ER stress, UPR activation, and apoptosis in mouse pancreatic acinar 266-6 cells [25]. In this alcohol-exposed in vitro model, the addition of exogeneous MANF ameliorated ER stress and cell death, whereas siRNA knockdown of MANF worsened cellular injury, suggesting a protective role of MANF in alcohol-induced pancreatic injury [25].

Pancreatic ductal cells are another key cell type in the exocrine pancreas and also depend on the functionality of ER homeostasis and UPR. Ductal cell alkaline secretions, regulated by the cystic fibrosis transmembrane conductance regulator (CFTR), play an important role in the pathophysiology of pancreatic disorders, as impaired pancreatic secretions and CFTR dysfunction have been observed in different forms of CP [26,27]. Alcohol exposure has been shown to cause ER dysfunction in pancreatic ductal cells through defective protein folding of CFTR at the ER in rodent models of alcoholic pancreatitis [28].

In this study, we sought to examine MANF expression in the pancreas of patients with chronic alcoholic and chronic non-alcoholic pancreatitis. Utilizing immunohistochemistry and digital image analysis, we found increased MANF expression in the ducts of chronic pancreatitis relative to normal tissue, a correlation between MANF ductal expression and the amount of fibrosis present, and reduced MANF expression in the acini of chronic alcoholic pancreatitis relative to non-alcoholic pancreatitis and the normal pancreas. These findings have implications for the role of MANF in the pathogenesis of chronic pancreatitis.

## 2. Materials and Methods

In this cross-sectional study, we used digital image analysis to quantitate the immunohistochemical expression of mesencephalic astrocyte-derived neurotrophic factor (MANF) in ductal and acinar cells of pancreata to compare the expression between patients diagnosed with chronic non-alcoholic pancreatitis (CNP) and those diagnosed with chronic alcoholic pancreatitis (CAP). This study was approved by the University of Iowa Biomedical Institutional Review Board and conducted under IRB-01; IRB ID #201112740 and utilized tissue from the pathology archives. The surgical pathology archives (from January 1971 to May 2021) and autopsy archives (from January 2018 to December 2020) of the University of Iowa Hospitals and Clinics (UIHC) Department of Pathology were searched for diagnoses containing various combinations of the following phrases: “pancreatitis”, “pseudocyst”, “IgG4”, “autoimmune pancreatitis”, “solid pseudopapillary”, “neuroendocrine”, and “alcohol”. Due to their relative rarity, all cases of autoimmune pancreatitis, non-tumor-associated obstructive pancreatitis, alcoholic pancreatitis, and hereditary pancreatitis were evaluated for study inclusion. Additionally, a cohort of sequentially accessioned tumor-associated obstructive pancreatitis was also evaluated for study inclusion. Cases excluded from further consideration included those with tissue blocks not housed within the UIHC archives and cases in which patients had received neoadjuvant therapy. The slides of candidate cases were retrieved and reviewed; cases with extensive autolysis, extensive necrosis, or extremely scant tissue were excluded from further consideration. A total of 92 cases were identified fulfilling these criteria. For each case, the patient’s age, gender, and alcohol consumption were retrieved from the electronic medical record during May of 2021. 

Cases with partially autolyzed or necrotic tissue (i.e., autopsy cases) or cases with scant tissue unsuitable for tissue microarray (TMA) construction were subsequently studied using whole-slide analysis. For the remainder of the cases, duplicate 1.5 mm diameter cores targeting tissue regions histologically representative of CAP and CNP were allocated among separate paraffin blocks to construct a total of five TMAs. A sixth TMA was similarly constructed targeting histologically normal pancreatitis. All blocks (TMA and whole slide) were cut as 5 µm sections and placed on glass slides. Immunohistochemistry (IHC) was performed on a DAKO Autostainer Link48 following heat-induced antigen retrieval using a DAKO PT Link. DAKO’s EnVision FLEX visualization system was used. The following antibody was assessed: MANF (abcam, clone ARMET/ARP; 1:400 dilution; high pH retrieval). Counterstaining was performed with Harris hematoxylin. H&E (hematoxylin and eosin)-stained TMA slides were reviewed, and cores were designated by author A.B. blinded to clinical history as being histologically representative of chronic pancreatitis (i.e., evidence of pancreatic parenchymal replacement with fibrous tissue) or normal pancreatic tissue (NPT). Cases were designated as CAP if the patient had a diagnosis of alcoholic pancreatitis recorded in their medical record or had heavy alcohol use (as defined by the National Institute on Alcohol Abuse and Alcoholism; >4 drinks on any day or >14 drinks per week for men and >3 drinks on any day or >7 drinks per week for women) plus histologic evidence of chronic pancreatitis.

IHC slides were digitally scanned (3DHISTECH P1000 Slide Scanner; Budapest, Hungary) and uploaded into the image analysis platform Halo (Indica Labs). For each case, a core annotation, an acini annotation, and a duct annotation were created by author N.C. blinded to clinical history. For the cases undergoing whole-slide analysis, a 1.5 mm diameter circular core annotation was manually created to best capture non-autolyzed tissues; for the cases represented using TMA, the entire core was annotated. The acini annotations were made within the core annotations and contained up to ten (if available) representative clusters of acinar cells. The duct annotations were also made within the core annotations and contained up to 10 (if available) representative, non-tangentially sectioned ducts; only ducts with a clear layer of surrounding connective tissue were annotated. Halo’s Area Quantification v2.1.11 algorithm was run in each core annotation and (since acinar parenchyma is strongly MANF expressing and occupies most of the pancreatic cross-sectional area) fibrosis was estimated as the percent non-DAB (diaminobenzidine)-positive tissue. Halo’s CytoNuclear v.2.0.9 algorithm was run on each of the acini and duct annotations and the average DAB positive cytoplasmic optical density (OD) for each tissue compartment was recorded. For cases which two cores of histologic chronic pancreatitis were sampled, the case OD was recorded as the average of the two core ODs. For cases which both a chronic pancreatitis core and a NPT core were sampled, each OD was recorded separately. 

Descriptive statistics, Mann–Whitney and Wilcoxon tests, and simple linear regressions were performed using GraphPad Prism 9.4.0. Multiple linear regression was performed using SPSS version 28; *p*-values of less than 0.05 were considered statistically significant.

## 3. Results

A total of 92 cases (14 autopsy, 78 surgical pathology) of chronic alcoholic pancreatitis (CAP) and chronic non-alcoholic pancreatitis (CNP) were identified and underwent mesencephalic astrocyte-derived neurotrophic factor (MANF) immunohistochemical (IHC) staining. Five cases (four autopsy of CAP, one surgical pathology of CNP) were excluded from statistical analysis following MANF IHC staining due to insufficient analyzable tissue present secondary to autolysis and tissue processing artifact. Of the 87 remaining cases, 72 were obtained following surgery for lesion removal, 10 were obtained from autopsy, 3 were obtained following surgery for recurrent pancreatitis, 1 case was obtained following surgery for trauma, and for the remaining case the indication for surgery was unknown (1994 case with limited available clinical history). Additional information regarding demographics, clinical information, and final histologic diagnoses is presented in Table 1.

A histologic review following tissue microarray (TMA) creation showed that tissue sampling from 15 cases yielded only CAP, 45 cases yielded only CNP, 14 cases yielded normal pancreatic tissue (NPT), and 13 cases yielded both CNP and normal tissue (due to core duplication during tissue microarray creation). In total, 73 cases demonstrated chronic alcoholic and non-alcoholic pancreatitis (CANP) (58 cases demonstrated CNP and 15 cases demonstrated CAP) and 27 cases demonstrated NPT. Additional information regarding the histologic cohorts is presented in Table 2.

Descriptive statistics of the digital image analysis results (stratified by gender) along with Mann–Whitney tests are shown in Table 3. The amount of fibrosis present in CANP exceeded that of NPT (70% versus 34%, respectively, *p* < 0.05). The MANF OD (optical density) of ducts in CANP exceeded that of NPT (0.19 versus 0.10, respectively, *p* < 0.05) and the MANF OD of ducts in CAP cases exceeded that of CNP cases (0.27 versus 0.17, respectively, *p* < 0.05). Both differences in MANF OD in ducts remained statistically significant among males when stratified by gender (*p* < 0.05 and *p* < 0.05, respectively), however, were not significant among females (*p* = 0.07 and *p* = 0.24, respectively). The MANF OD in acini in CNP cases exceed that of CAP cases (1.04 with versus 0.81, respectively, *p* < 0.05). Dot plots with Mann–Whitney results along with representative microphotographs are displayed in Figure 1, Figure 2 and Figure 3.

Simple linear regression showed a statistically significant correlation between the amount of fibrosis versus duct OD (y = 0.002499x + 0.0160; 95% confidence interval of slope 0.001404 to 0.003594; *p* < 0.05, R^2^ = 0.18). No correlation was noted between acini OD versus duct OD (*p* = 0.13) or the amount of fibrosis versus acini OD (*p* = 0.81). The results of these regression studies are shown in Figure 4.

Multiple linear regression was used to test if age, sex, etiology, acinar OD, and duct OD predicted fibrosis in chronic pancreatitis. The overall regression was very nearly statistically significant (R^2^ = 0.17, F (5, 59) = 2.5, *p* = 0.052). It was found that duct OD predicted fibrosis (β = 0.37, *p* = 0.009), while none of the other independent variables did.

Wilcoxon testing of the 13 cases with representation of both CNP and NPT showed statistically significant differences in fibrosis present within the matched pairs (*p* < 0.05), as shown in Figure 5, however, no statistically significant differences were seen between acini and duct OD.

## 4. Discussion

Alcoholic pancreatitis is the most common cause of chronic pancreatitis (CP), yet the pathophysiological mechanism of alcohol’s involvement in CP remains to be completely elucidated [4,29,30]. This study aimed to investigate the expression of mesencephalic astrocyte-derived neurotrophic factor (MANF), a critical factor regulating endoplasmic reticulum (ER) homeostasis, in the pancreas of patients with chronic alcoholic pancreatitis (CAP) and chronic non-alcoholic pancreatitis (CNP). We demonstrated that MANF expression was higher in ducts of CAP than CNP. In contrast, MANF expression in acini was lower in CAP than CNP and normal pancreatic tissue (NPT). There was also a positive correlation between fibrosis and MANF levels in the ducts.

ER stress has been proposed as an important cellular and molecular mechanism underlying acinar cell injury in pancreatitis [11,20,25,31]. Sustained ER stress and unfolded protein response (UPR) activation have been observed in pancreatic acini in human CP [31]. ER stress can either directly cause pancreatic injury or sensitize the pancreas to other pathologic insults, such as alcohol [20,32,33,34,35,36]. In experimental models of alcoholic pancreatitis, chronic alcohol exposure resulted in ER stress, and reduced levels of UPR regulators such as Xbp1s or Mist1 worsen alcohol-induced damage to the pancreas [10,37]. As a UPR regulator, MANF has been shown to play a pivotal role in regulating the three arms of UPR signaling (i.e., IRE1α, PERK, and ATF6α) and ER-stress-induced acinar cell death [11,25]. Knockdown of MANF exacerbates alcohol-induced damage to the mouse pancreatic 266-6 acinar cells, whereas overexpression of MANF attenuated alcohol-induced ER stress and cellular death, suggesting a protective role of MANF in alcohol-induced pancreatic injury [25]. 

In this study, we found that in vivo expression levels of MANF in pancreatic acini of CAP cases were significantly lower than that of CNP or normal cases. Moreover, there was no significant difference in the acinar MANF levels between CNP and that of normal cases. Taken together, these results suggest that low acinar MANF levels may be a co-factor for the development of alcoholic CP; alcohol exposure may reduce acinar MANF levels, which makes acinar cells more susceptible to ER-stress-associated acinar injuries. It is also possible that the lower acinar MANF levels exacerbate alcoholic pancreatitis, which is consistent with our previous findings in a cell culture model [25]. There was no correlation in the amount of fibrosis versus MANF level in pancreatic acini, suggesting that although low acinar MANF level may contribute to the alcoholic CP, it does not relate to the severity of fibrosis. 

To our knowledge, the function of MANF in pancreatic ductal cells has never been studied. Although research on the involvement of pancreatic ductal cells in CP is limited, there are two ductal ion channel proteins, CFTR and CLDN2, that have been shown to be involved in alcoholic CP [28,38]. Unpublished data from our laboratory indicate that MANF is a suppressor of CLDN2 in the mouse pancreas. Here we have found that the expression levels of MANF in pancreatic ductal cells of CANP cases were higher than that of NPT; we also observed higher ductal MANF levels in CAP cases than that of CNP cases. Both differences in ductal MANF levels remained statistically significant among males but not in females. Furthermore, our study demonstrated an association between the amount of fibrosis and ductal MANF levels. These results demonstrate an association between ductal MANF levels and CP, especially CAP in men. It appears that MANF expression shows a cell-type-specific response to alcohol exposure. For example, it has been reported that alcohol exposure causes neuronal cell death and upregulated MANF expression in the brain [17,39]. It is unclear how high ductal MANF levels are involved in the pathology of alcoholic CP. It is possible that MANF is induced to repair alcohol-induced damage to pancreatic ductal cells. Further experiments are needed to test whether high ductal MANF levels suppress CLDN2 expression leading to dysfunction of ductal cells in human CP.

Recent studies suggest that MANF can regulate pancreatic ER homeostasis and β cell functions in humans and animals [13,40,41]. For example, recombinant MANF promotes β cell proliferation and survival in vitro [13]. MANF-deficient mice display ER stress in the pancreas and develop diabetes due to progressive postnatal reduction in β cell mass [11]. Increased MANF concentrations in serum are associated with the clinical manifestation of type 1 diabetes in children [40]. Thus, it appears that MANF is not only involved in maintaining ER homeostasis but also in regulating pancreatic function. Although the role of MANF in the endocrine system of the pancreas, such as β cell functions, has been established, the effects of MANF on the exocrine compartment of the pancreas are less studied and there is little information. Particularly, the role of MANF in pancreatic acinar cells and ductal cells is unknown. Since pancreatitis is initiated in acinar cells and ER stress plays a critical role during the pathogenesis of pancreatitis, understanding the role of MANF in acinar cells in response to alcohol exposure is important, and our study for the first time demonstrates that MANF expression is altered in acinar and ductal cells of CAP. Thus, the results obtained from this clinical study are consistent with our previous in vitro research that showed that MANF in acinar cells was responsive to alcohol exposure and involved in cell survival and ER stress following alcohol exposure. Although heavy drinking is a risk factor, only a small portion of heavy drinkers develop pancreatitis [42]. Therefore, in response to alcohol exposure, the pancreas may initiate compensatory responses, and ER stress or UPR is one of them. In our recent review article [11], we proposed that the interaction of alcohol exposure and ER stress is involved in the pathogenesis of alcoholic pancreatitis through the following mechanisms: (1) Since alcohol exposure causes ER stress in the pancreas, a pre-existing imbalance of ER homeostasis or ER dysfunction may exacerbate alcohol-induced ER stress. This is beyond UPR’s ability to restore and ultimately results in severe pancreatic damages and pancreatitis. (2) The genetic mutations or protein alterations in key components of UPR or ER-associated degradation (ERAD) pathways may already impair pancreatic cells’ ability to alleviate ER stress. Upon alcohol exposure, sustained and severe ER stress results in cell death, inflammation, and other pancreatic damages. (3) Additionally, alcohol exposure, especially chronic and heavy alcohol consumption, may disrupt ER homeostasis or impair UPR or ERAD systems, sensitizing pancreatic cells to other genetic or environmental stressors. As a result, alcohol abusers are more susceptible to etiological initiators of pancreatitis. The current study supports the hypothesis and suggests that MANF plays an important role in the pathogenesis of CAP.

In conclusion, our study demonstrates a cell-type-specific expression pattern of MANF in CP with lower acinar MANF levels observed in CAP cases and higher ductal MANF levels associated with CAP and increased fibrosis. Further studies using human samples and experimental models are necessary to elucidate MANF’s role in alcoholic pancreatitis. For example, it would be interesting to compare serum levels of MANF between CAP and CNP to determine whether serum MANF may be indicative of pancreatic damage. To determine the exact role of MANF in the pathogenesis of alcoholic pancreatitis, it would also be important to establish new experimental models, such as acinar-cell-specific or ductal-cell-specific MANF knockout animals or cell lines to determine whether loss of MANF exacerbates alcohol-induced pancreatic injury. To associate the role of MANF to ER stress in the context of CAP, it is necessary to further examine ER stress and the expression of UPR proteins in the acinar and ductal cells in which MANF expression is altered.

## Figures and Tables

**Figure 1 biomedicines-11-00434-f001:**
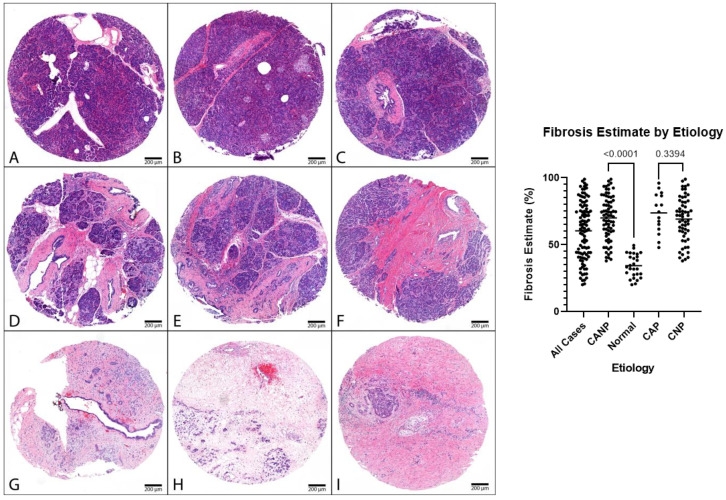
Representative cohort cases, H&E. Left: (**A**–**C**) Representative normal pancreatic tissue (NPT) with fibrosis estimates of 23%, 35%, and 44%, respectively (NPT cohort average of 34%). (**D**–**F**) Representative chronic pancreatitis tissue with fibrosis estimates of 61%, 65%, and 70%, respectively. (**G**–**I**) Representative chronic pancreatitis tissues with fibrosis estimates of 92%, 95%, and 96%, respectively. Panels (**D**,**F**,**H**) are cases of CNP (CNP cohort average of 69%). Panels (**E**,**G**,**I**) are cases of CAP (CAP cohort average of 74%): 40× magnification. (**Right**): Dot plot of fibrosis estimates, combined genders, with *p*-values displayed. H&E, hematoxylin and eosin.

**Figure 2 biomedicines-11-00434-f002:**
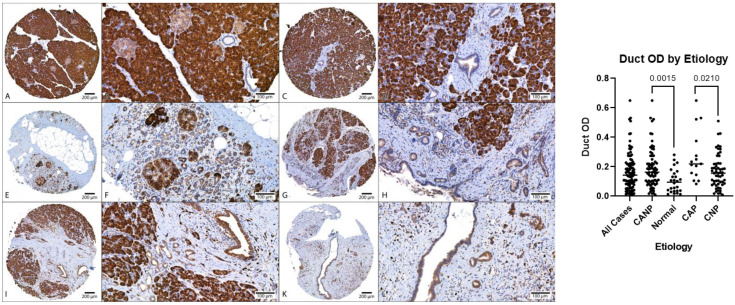
Comparison of MANF duct OD, DAB with hematoxylin counterstain. Left: (**A**–**D**) Representative normal pancreatic tissue (NPT) with duct OD averages of 0.10 and 0.06 (NPT average of 0.10). (**E**–**H**) Representative chronic non-alcoholic pancreatitis (CNP) with duct OD averages of 0.14 and 0.17, respectively (CNP cohort average of 0.17). (**I**–**L**) Representative chronic alcoholic pancreatitis (CAP) with duct OD averages of 0.38 and 0.25, respectively (CAP cohort average of 0.27). (**A**,**C**,**E**,**G**,**I**,**K**) at 40× magnification; (**B**,**D**,**F**,**H**,**J**,**L**) at 200× magnification. (**Right**): Dot plot of duct OD estimates, combined genders, with *p*-values displayed. OD, optical density; DAB, diaminobenzidine.

**Figure 3 biomedicines-11-00434-f003:**
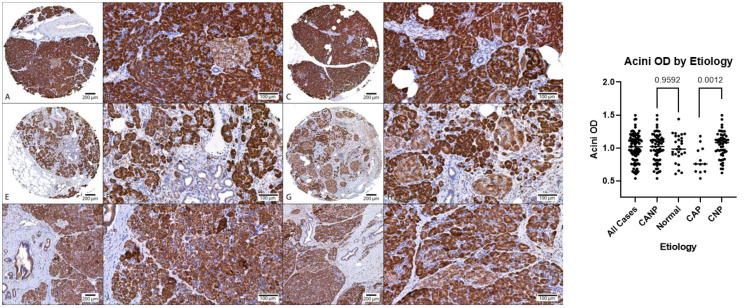
Comparison of MANF acini OD, DAB with hematoxylin counterstain. (**A**–**D**) Representative normal pancreatic tissue (NPT) with acini OD averages of 0.94 and 1.11 (NPT average of 1.00). (**E**–**H**) Representative chronic non-alcoholic pancreatitis (CNP) with acini OD averages of 1.02 and 0.96, respectively (CNP cohort average of 1.04). (**I**–**L**) Representative chronic alcoholic pancreatitis (CAP) with acini OD averages of 0.75 and 0.81, respectively (CAP cohort average of 0.81). (**A**,**C**,**E**,**G**,**I**,**K**) at 40× magnification; (**B**,**D**,**F**,**H**,**J**,**L**) at 200× magnification. (**Right**): Dot plot of duct OD estimates, combined genders, with *p*-values displayed. OD, optical density; DAB, diaminobenzidine.

**Figure 4 biomedicines-11-00434-f004:**
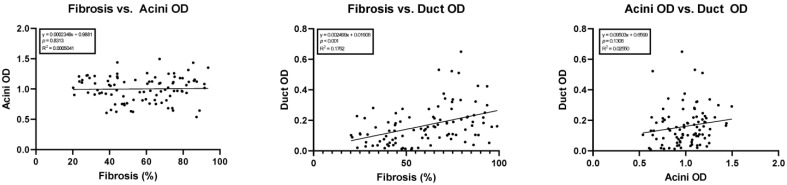
Linear regressions comparing cohort fibrosis, acini optical density, and ductal optical density. OD, optical density.

**Figure 5 biomedicines-11-00434-f005:**
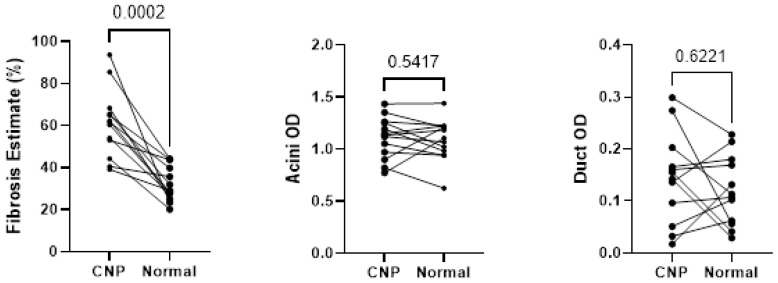
Wilcoxon testing comparing fibrosis, acini optical density, and duct optical density between histologic designations of cases with both normal pancreatic tissue and chronic non-alcoholic pancreatitis tissues sampled. CNP, chronic non-alcoholic pancreatitis; OD, optical density.

**Table 1 biomedicines-11-00434-t001:** Patient demographics, clinical history, histologic diagnosis. M, male; F, female.

Accession Type *Accession Reason*	#	Pathologic Diagnosis	Age (Average, (Range))	Sex
**Total**	**87**		**58 (11–82)**	**42M/45F**
**Surgical Pathology**	**77**	**Primary Histologic Diagnosis**	**58 (11–82)**	**36M/41F**
** *Lesion Removal* **	72	30 ductal adenocarcinoma 15 intraductal papillary mucinous neoplasm 7 neuroendocrine neoplasm 5 pancreatic pseudocyst 4 solid pseudopapillary tumor 3 pancreatic intraepithelial neoplasia 2 ampullary adenocarcinoma 2 mucinous cystic neoplasm 1 benign non-neoplastic cyst 1 benign common bile duct stenosis 1 lymphoepithelial cyst 1 metastatic cancer	60 (11–82)	33M/39F
** *Recurrent Pancreatitis* **	3	1 obstructive choleliths 1 cystic fibrosis 1 alcoholic pancreatitis	43 (23–60)	2M/1F
** *Trauma* **	1	1 splenic hematoma	53 (53)	1M
** *Unsure* **	1	1 unsure	47 (47)	1F
**Autopsy**	**10**	**Cause of Death** 6 complications of chronic alcoholism 2 pyelonephritis 1 acute pancreatitis (non-alcoholic) 1 bronchopneumonia	**43 (25–59)**	**6M/4F**

**Table 2 biomedicines-11-00434-t002:** Study cohort. CANP, chronic alcoholic and non-alcoholic pancreatitis; CAP, chronic alcoholic pancreatitis; CNP, chronic non-alcoholic pancreatitis; F, female; M, male; NPT, normal pancreatic tissue.

Tissue Demonstrated (by case)	#	Accession Reason	Age (Average, (Range))	Sex
**CAP only**	15	9 autopsy 3 lesion removal 1 recurrent pancreatitis 1 trauma 1 unsure	48 (28–68)	10M/5F
**CNP only**	45	43 lesion removal 1 autopsy 1 recurrent pancreatitis	62 (17–82)	22M/23F
**NPT only**	14	14 lesion removal	52 (19–75)	5M/9F
**Both CNP and NPT**	13	12 lesion removal 1 recurrent pancreatitis	58 (11–80)	5M/8F
**Tissue Demonstrated (total)**	#	**Accession Reason**	**Age (Average, (Range))**	**Sex**
**CANP**	73	58 lesion removal 10 autopsy 3 recurrent pancreatitis 1 trauma 1 unsure	59 (11–82)	37M/36F
**CAP**	15	9 autopsy 3 lesion removal 1 recurrent pancreatitis 1 trauma 1 unsure	48 (28–68)	10M/5F
**CNP**	58	55 lesion removal 2 recurrent pancreatitis 1 autopsy	61 (11–82)	27M/31F
**NPT**	27	26 lesion removal 1 recurrent	55 (11–80)	10M/17F

**Table 3 biomedicines-11-00434-t003:** Results of statistical analysis. CANP, chronic alcoholic and non-alcoholic pancreatitis; CAP, chronic alcoholic pancreatitis; CNP, chronic non-alcoholic pancreatitis; OD, optical density.

Fibrosis Estimate						
	**N**	**Mean (%)**	**Range (%)**	***p*-Value**		
				**Both**	**Male**	**Female**
All Cases	100	60	(20–99)			
Male	47	59	(23–99)			
Female	53	70	(20–95)			

CANP	73	70	(38–99)			
Male	37	70	(38–99)			
Female	36	71	(44–95)			
				<0.05	<0.05	<0.05
Normal	27	34	(20–50)			
Male	10	36	(23–48)			
Female	17	34	(20–50)			

CAP	15	74	(48–96)			
Male	10	76	(48–96)			
Female	5	70	(52–87)			
				0.34	0.24	0.86
CNP	58	69	(38–99)			
Male	27	67	(38–99)			
Female	31	71	(44–95)			

**Duct OD**						
	**N**	**Mean**	**Range**	***p*-value**		
				**Both**	**Male**	**Female**
All Cases	98	0.17	(0.01–0.65)			
Male	45	0.17	(0.01–0.52)			
Female	53	0.16	(0.01–0.65)			

CANP	72	0.19	(0.01–0.65)			
Male	36	0.19	(0.02–0.52)			
Female	36	0.19	(0.01–0.65)			
				<0.05	<0.05	0.07
Normal	26	0.1	(0.01–0.28)			
Male	9	0.09	(0.01–0.25)			
Female	17	0.11	(0.01–0.28)			

**Duct OD**						
	**N**	**Mean**	**Range**	***p*-value**		
				**Both**	**Male**	**Female**
CAP	15	0.27	(0.08–0.65)			
Male	10	0.25	(0.10–0.52)			
Female	5	0.32	(0.08–0.65)			
				<0.05	<0.05	0.24
CNP	57	0.17	(0.01–0.51)			
Male	26	0.17	(0.02–0.43)			
Female	31	0.17	(0.01–0.51)			

**Acini OD**						
	**N**	**Mean**	**Range**	***p*-value**		
				**Both**	**Male**	**Female**
All Cases	92	1	(0.54–1.50)			
Male	41	1.01	(0.64–1.50)			
Female	51	1	(0.54–1.44)			

CANP	65	1	(0.54–1.50)			
Male	31	0.99	(0.64–1.50)			
Female	34	1.01	(0.54–1.43)			
				0.96	0.46	0.51
Normal	27	1	(0.61–1.44)			
Male	10	1.04	(0.75–1.23)			
Female	17	0.98	(0.61–1.44)			

CAP	12	0.81	(0.54–1.18)			
Male	7	0.83	(0.64–1.18)			
Female	5	0.8	(0.54–1.10)			
				<0.05	<0.05	<0.05
CNP	53	1.05	(0.62–1.50)			
Male	24	1.04	(0.68–1.50)			
Female	29	1.05	(0.62–1.43)			

## Data Availability

Data will be available if requested.

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
