# Peer review of "Altered MANF Expression in Pancreatic Acinar and Ductal Cells in Chronic Alcoholic Pancreatitis: A Cross-Sectional Study"

_biomedicines, 2023, doi:10.3390/biomedicines11020434_

Round 1
Reviewer 1 Report
Chronic pancreatitis is a progressive inflammatory disorder of the pancreas that adversely affects quality of life due to irreversible injury to the pancreas. Alcohol abuse is the most common risk factor for chronic pancreatitis. However, the pathogenesis of alcoholic chronic pancreatitis remains incompletely understood. Thus, the aim of this study was to investigate the expression of mesencephalic astrocyte-derived neurotrophic factor, a critical factor regulating endoplasmic reticulum homeostasis, in the pancreas of patients with chronic alcoholic pancreatitis. The authors found that in vivo expression levels of mesencephalic astrocyte-derived neurotrophic factor in pancreatic duct of chronic alcoholic pancreatitis cases were higher than that of chronic non-alcoholic pancreatitis cases. While the expression of mesencephalic astrocyte-derived neurotrophic factor in acini of chronic alcoholic pancreatitis cases were lower than that of chronic non-alcoholic pancreatitis cases. The authors conclude that further studies using human samples and experimental models are necessary to elucidate role of mesencephalic astrocyte-derived neurotrophic factor in alcoholic pancreatitis. The manuscript is well-written and the methods sound. I did not have any major concerns, only minor issues listed below.
Please add scale bars in Figure 1, 2, and 3.
Author Response
We thank the reviewers for their careful evaluation of our work. In this revised manuscript, we have revised the manuscript to address their comments. The point-to-point responses are provided as below. Changes in the revised manuscript are tracked or underlined.
Reviewer #1:
Minor points: “Please add scale bars in Figure 1, 2, and 3”
Response: We have added scale bars to referred figures.
Reviewer 2 Report
The paper is well written and it focus on one interesting topic.
I have very minimal comments and suggestions:
1) the paper is an observational study. It should be cited into the title as cross sectional study.
2) please follow the strobe guidelines for writing each section
3) This study shows that there is a positive correlation between fibrosis and MANF levels in the ducts but not OR are reported. I suggest a multivariate analysis including all risk factors.
Author Response
We thank the reviewers for their careful evaluation of our work. In this revised manuscript, we have revised the manuscript to address their comments. The point-to-point responses are provided as below. Changes in the revised manuscript are tracked or underlined.
Reviewer #2:
Minor point: “The paper is well written, and it focus on one interesting topic. I have very minimal comments and suggestions:
1) the paper is an observational study. It should be cited into the title as cross sectional study.
2) please follow the strobe guidelines for writing each section.
3) This study shows that there is a positive correlation between fibrosis and MANF levels in the ducts but not OR are reported. I suggest a multivariate analysis including all risk factors.
Response:
- We have revised the tile in response to the suggestion.
- We have modified our writing in each section to be consistent with the strobe guidelines.
- We have run a multivariate analysis including all risk factors on fibrosis and MANF expression. The results are included.